# Application of a Semi-Empirical Dynamic Model to Forecast the Propagation of the COVID-19 Epidemics in Spain

**Juan Carlos Mora** [1,*] **, Sandra Pérez** [2] **and Alla Dvorzhak** [1]

[1] Department of Environment, CIEMAT, Avenida Complutense, 40, 28040 Madrid, Spain; alla.dvorzhak@ciemat.es

[2] Sercomex Pharma, C/ Pollensa, 2, 28232 Las Rozas de Madrid, Spain; sandra.perez@sercomex.es

[*] Correspondence: jc.mora@ciemat.es; Tel.: +34-91-346-6751

**Abstract:** A semiempirical model, based in the logistic map, was developed to forecast the different phases of the COVID-19 epidemic. This paper shows the mathematical model and a proposal for its calibration. Specific results are shown for Spain. Four phases were considered: non-controlled evolution; total lock-down; partial easing of the lock-down; and a phased lock-down easing. For no control the model predicted the infection of a 25% of the Spanish population, 1 million would need intensive care and 700,000 direct deaths. For total lock-down the model predicted 194,000 symptomatic infected, 85,700 hospitalized, 8600 patients needing an Intensive Care Unit (ICU) and 19,500 deaths. The peak was predicted between the 29 March/3 April. For the third phase, with a daily rate $r = 1.03$, the model predicted 400,000 infections and $46,000 \pm 15,000$ deaths. The real $r$ was below 1%, and a revision with updated parameters provided a prediction of 250,000 infected and $29,000 \pm 15,000$ deaths. The reported values by the end of May were 282,870 infected and 28,552 deaths. After easing of the lock-down the model predicted that the health system would not saturate if $r$ was kept below 1.02. This model provided good accuracy during epidemics development.

**Keywords:** semi-empirical model; logistic map; COVID-19; SARS-CoV-2

## 1. Introduction

A new respiratory disease, initially dominated by pneumonia, and caused by a coronavirus, was detected at the province of Hubei, in China, at the end of 2019. It was initially named by the World Health Organization (WHO) as 2019-nCoV [1] and renamed in February 2020 by the International Committee on Virus Taxonomy as Severe Acute Respiratory Syndrome coronavirus 2 (SARS-CoV-2), recognizing it as a sister of the SARS-CoV viruses [2,3]. The same day the WHO [4] named the disease as Coronavirus Disease 2019 (COVID-19).

Many efforts have been made since then to mathematically model the spread of the disease in the whole world and in the different countries where the infection arrived. Modelling the epidemics has many practical uses: preparation of national health systems; make provisions of the necessary sanitary material; predict whether and when a saturation of the health system could occur; when and to what extent Non Pharmaceutical Interventions (NPI) [5] should be applied; predict the day when those countermeasures can be relaxed, and so forth. These theoretical approaches to predict the evolution of epidemics often use compartment models as simple as the SIR model (Susceptible, Infectious and Recovered—sometimes called Removed) [6], but this model can be increased in complexity to include different characteristics of an infectious epidemics. For example the model can include individuals who can infect others, without presenting symptoms, what is known as the SEIR model (Susceptible,

Exposed, Infectious and Removed); the model can also assume that people who have recovered from the disease lose the immunity after a given time, and therefore they could be infected again, giving rise to the SEIRS models (Susceptible, Exposed, Infectious, Removed and Susceptible); also the deaths and births can be included for long term epidemics, as is the case in the influenza; and many times compartments to distinguish deaths, recovered, hospitalized, and other situations, are included by using empirical ratios (see for instance References [7,8] for further information).

As in any consolidated scientific field, there is abundant literature describing the different mathematical models which can be applied to different diseases, for different population behaviour, or even to define the optimal control strategies [9–13]. In Reference [14] a review of the models used in the India to forecast the behaviour of the COVID-19 was carried out, including among others: compartments (SIR type) models, ARIMA models, machine learning based approaches and logistic curves. This review remarks the very important discrepancies found between those models in the predictions after lockdowns were applied.

Since the SARS-CoV-2 outbreak many efforts have been carried out to adapt these SIR type compartment models to the behaviour of this particular virus. For example, a conceptual representation of a compartment model for COVID-19 disease's spread, developed by the authors of this paper, is shown in Figure 1, where immunization of recovered is assumed to be lost after a given period of time, as happen in other infectious diseases (typically immunity is lost after less than 12 months in the case of the coronavirus causing common cold).

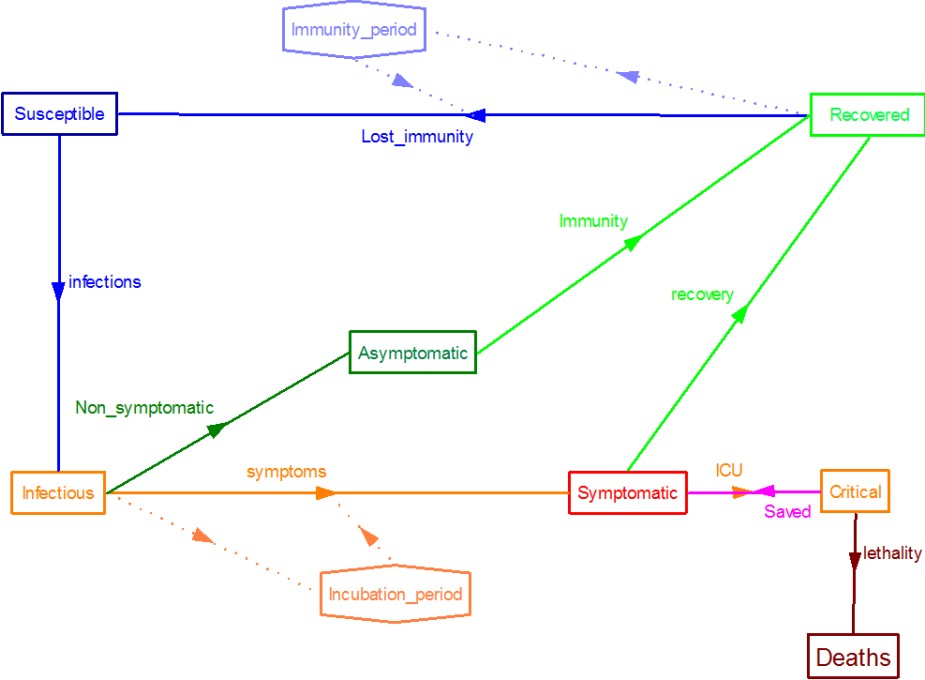

**Figure 1.** Example of a Susceptible, Infectious and Recovered (SIR) type compartment model adapted to simulate the behaviour of SARS-CoV-2. In this case we assumed that immunity would be lost in a given period of time.

Due to the difficulty of developing and calibrating these compartment models at the early stages of an epidemics, wrong conclusions are often reached, for instance predicting the timing of the epidemic, and many times the uncertainties associated with the results of the models are so wide that are not well accepted by the public. Sometimes the authors of such predictions blamed the quality of the statistical data [15,16]. However, this quality is severely affected by the urgency of the epidemic and could not probably be avoided in this or future outbreaks of epidemics. The continuous publications

of medical and epidemiological studies on the COVID-19, and the associated virus, don't make it easy to extract good quality information to adapt the models. But it must be accepted that this situation will be always the case—or even worse—when new diseases appear.

Some other problems associated with future predictions of the COVID-19 behaviour, like the influence of ambient temperature or humidity in the seasonality of the disease [17], are still unsolved. In the early stages many aspects of the behaviour of the SARS-CoV-2 virus has been associated with previous studies on similar viruses as the SARS virus, as the resilience in fomites [18] or the immunization of patients recovered from it [19]. Many other aspects which would also affect the choice of the best model are still under investigation, for example the possible lost of the immunity after some months, as happens with other human coronaviruses which cause other diseases like the 15% of the common cold cases [20].

During the outbreak of the epidemic in Spain several models were tested and a follow-up of the published results were performed to support Spanish national authorities in the decision-making process [21], producing a preliminary work covering all the phases which was published as a preprint [22]. The best results were obtained by using a semi-empirical approach presented herein, which has the advantage of performing accurate predictions with the minimum amount of information available during this epidemic which, very likely, will be the situation in future outbreaks.

This paper presents the mathematical development for a proposed semi-empirical model and the results obtained using it, focusing the results into the Spanish case. Some results obtained for other countries are also presented.

## 2. Materials and Methods

The semi-empirical model presented in this paper, with a proper calibration, produces accurate results at every stage of the epidemic: during the first spread of the disease, after the application of NPI (specifically total lock-down) which were applied in many countries, and after the ease of the lock-down.

Although the instant reproduction number (Rt) used by the epidemiologists for estimating the severity and evolution of an epidemic [23] is not used in this model, the basic reproduction number (R0) was derived for 10 different countries, ranging from 2.0 to 9.3, which is in a good agreement with previous estimations [24]. The R0 derived from real data are found in Table 1, giving an average of $5.8 \pm 2.4$, a value almost doubling early estimations [25].

**Table 1.** Basic Reproductive Number calculated for some studied countries.

| Country | R0 |
|---|---|
| France | 9.3 |
| USA | 8.2 |
| Slovenia | 7.5 |
| Norway | 6.9 |
| Italy | 6.7 |
| UK | 6.7 |
| Spain | 4.6 |
| Belgium | 3.5 |
| South Korea | 2.8 |
| Germany | 2.0 |

The model proposed in this paper applies the well-known logistic map, often used for describing the growth of populations and mentioned as an example of chaotic behaviour. This chaotic behaviour depends on a single parameter $r$ (Figure 2 shows a fractal created with the logistic map as a function of $r$). In this equation values of $r < 1$ would make an epidemic to extinguish. Any $r$ greater than 1 but below 3 will provide an equilibrium in the size of the population for the long term, while values of $r$ higher than 3.56995 would produce a chaotic behaviour on the size of the population.

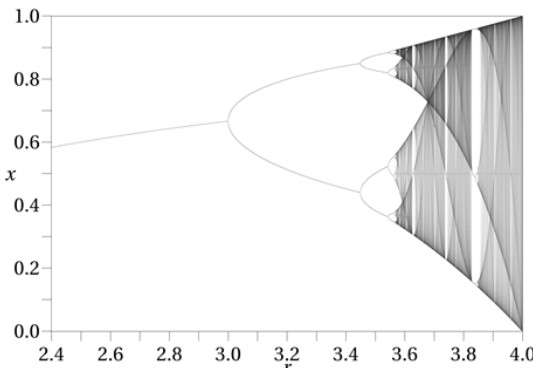

**Figure 2.** Bifurcation diagram for the logistic map as a function of r.

Therefore to determine the number of infected diagnosed cases Equation (1) is used:

$$I(t) = r \cdot \left( 1 - \frac{I(t-1)}{N} \right) \cdot I(t-1), \tag{1}$$

where $I(t)$ is the number of infected diagnosed cases at day $t$, $I(t-1)$ the infected diagnosed cases of the precedent day $t-1$, $r$ is the growth parameter of the logistic map (named hereafter daily infection rate), and $N$ the number of individuals susceptible to be infected (in Figure 2 a simplified example of this function, with constant parameters, is shown). It should be noted that the number of susceptible individuals used here is not necessarily the same as the number used for modellers applying SIR type models. The sub-index n will be used below to indicate the n-th day after the outbreak.

The behaviour of this function gives rise to the logistic function and the typical sigmoid shape of its cumulative distribution if $r < 2$, while it shows a chaotic behaviour if $r > 3.56995$ (see Figure 3). Other authors have studied the behaviour of this logistic function applied to the COVID-19 epidemics [26,27].

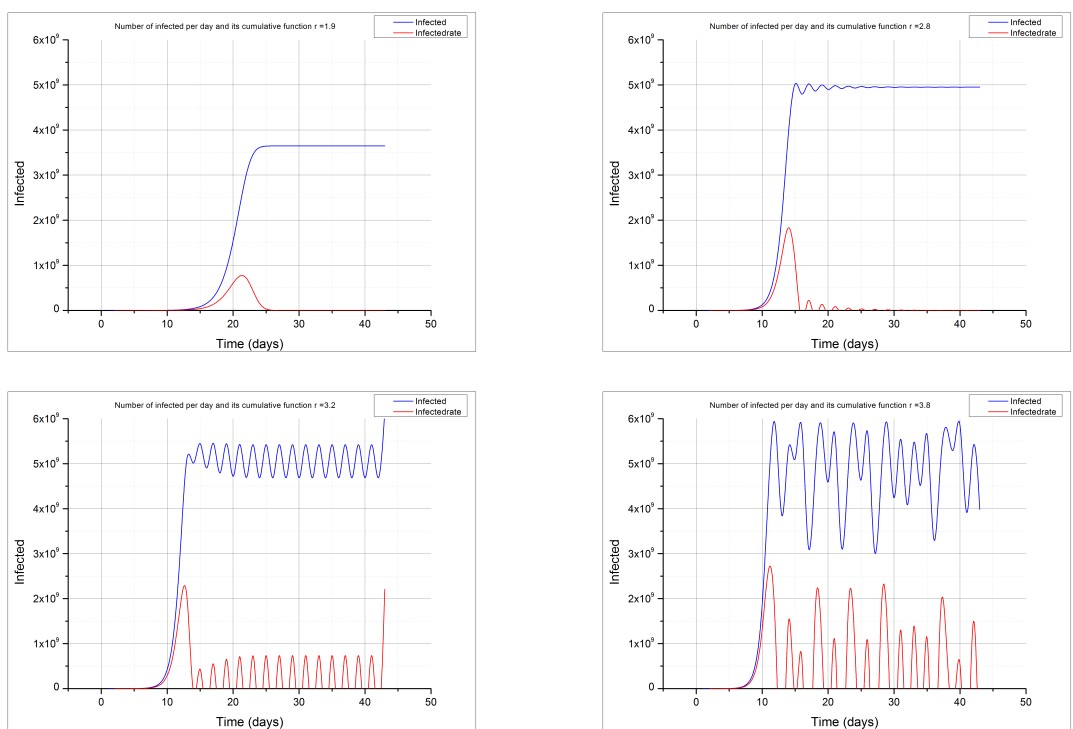

**Figure 3.** Number of infected obtained for the logistic map as a function of $r$ for the different options, from $r = 1.9$ to $r = 3.8$.

In order to compare with the values of the basic reproduction number in Table 2 the empirically determined values of the growth parameter $r$ are shown for the same countries in Table 1. This $r$ parameter is simply measured by dividing the new infected in a given day by the infected in the previous day. To avoid statistics biases $r$ was taken for each country as an average for the first 7 days after the initial detections of infected at each country. The $r$ in these countries was equal to $1.9 \pm 0.5$, ranging from 1.3 to 3.0. In this approach a value of $r < 3$ implies that, in absence of countermeasures, and independently of the initial value $I(0)$ there would be reached an unique equilibrium on the number of infected: $I(\infty) = N \cdot \frac{r-1}{r}$. Worldwide, in average, an equilibrium value of 3.65 billions of infected would be reached applying $r = 1.9$ and $N = 7.7 \times 10^9$ to the equation. The equilibrium values which would be reached, if no intervention was applied for each country, are shown at the Table 2.

**Table 2.** Growth parameter $r$ for the logistic map, empirically calculated for the same countries during the first days of the spread of the COVID-19 epidemics, and equilibrium value for the infected people if no countermeasures were applied.

| Country | $r$ | $I(\infty)$ (Millions) |
|---|---|---|
| France | 1.8 | 29.7 |
| USA | 1.3 | 75.7 |
| Slovenia | 2.2 | 1.1 |
| Norway | 2.0 | 2.7 |
| Italy | 3.0 | 40.2 |
| UK | 1.4 | 19.0 |
| Spain | 1.5 | 15.5 |
| Belgium | 1.7 | 3.8 |
| South Korea | 2.0 | 51.6 |
| Germany | 1.8 | 36.9 |

Therefore the basic quantity used to make predictions is the number of infected $I(t)$ reported by each country or region. This model does not need considering asymptomatic infected or questions what is the real number of infected, but makes use of the data reported. However, as demonstrated in the case of the "Diamond Princess" cruise, nearly the 70% of the infected would be asymptomatic and undiagnosed [28].

Other quantities needed to provide advice to the authorities are the number of inpatients who would need medical attention at the hospital ($H_n$), the number of those who would need intensive care ($C_n$) and the number of deaths ($D_n$), all of them at each time $t$. $H_n$ and $D_n$ are calculated as a fraction of the number of the diagnosed infected cases at time t ($I_n$), and $C_n$ as fraction of $H_n$. Obviously the number of recovered patients ($R_n$) is given by the fraction $(1 - D_n)$. The fraction used to calculate $D_n$ in this way is referred to as the case fatality rate (CFR), determined as $CFR = \frac{D_n}{D_n + R_n}$. This is proved to be more practical during the outbreak than other approaches as the mortality rate for the whole population which can be only experimentally known at the end of the epidemic.

A delay must be included to represent the time elapsed between a death and its report to the authorities, including the time needed to perform the diagnoses (usually by using the polymerase chain reaction (PCR) technique).

All of these numbers are crucial to policy makers in order to take well founded decisions. However, to perform reliable predictions an appropriate calibration of the model is needed which will depend on the specific situation of each phase of the epidemics.

*2.1. Initial Parametrization*

All the parameters of the model are empirically calibrated by averaging the available information in a studied region. This calibration is feasible at the early stages, when the data available cover only few days, but it can also be dynamically adjusted during the whole evolution of the epidemics.

For performing reliable predictions at the very beginning of the outbreak, the information from previously infected countries can be used as initial calibration of the model.

SARS-CoV-2 is assumed to infect with the same probability to every human, disregarding sex or age. Being a new human virus, no immunization was previously acquired, by natural or artificial (vaccine) means. For that reason the number of people which can be infected, $N$, was initially assumed to be the whole population of the studied region, whatever the size of that region is. For the sake of simplicity the total population of a country (or a region, as are the so-called autonomous communities in Spain) is initially used. In the case of Spain the total population $N = 46.6 \times 10^6$ was initially used.

The daily infection rate, $r$, can be dynamically determined - using all the data collected to average a given period - dividing the number of infected the day $n + 1$ ($I_{n+1}$) by the number of infected at day $n$ ($I_n$). For Spain, averaging the daily infection rate during the first 7 days of the outbreak, from the 26 February/3 March, an $r = 1.5$ was obtained. Following the same method $r$ values were obtained for 10 countries (see Table 2). This parameter however with the actions taken by the governments and the population, as the social distancing, the frequent hands washing or the use of masks.

The fraction of the infected who need to be hospitalized ($H = \frac{H_n}{I_n}$) is dynamically determined using the data acquired at each region (or state, or country), averaged for the whole period since the beginning of the epidemic. The same was done for the fraction of inpatients needing an ICU ($C = \frac{C_n}{H_n}$). An initial factor of patients needing an Intensive Care Unit (ICU) from the number of infected cases $H \cdot C = 0.05$ to $0.15$ was computed from the studies in Eastern countries.

As the reported data for the infected patients were given as accumulated since the beginning of the epidemic, all the other quantities were also obtained as cumulative functions. Due to its special configuration of the health system, this was a problem in Spain, as different regions decided to report different quantities and then initially was impossible to obtain accumulated data for hospitalized persons, or patients which needed an ICU for the country. Daily rates were interesting to calculate, for instance, the day where the maximum of infections or deaths (peak of the curve) would occur.

For the CFR, the value measured near the equilibrium in China was used ($CFR = 0.0578$, as measured the 4 March—see Figure 4 to see the evolution of the CFR in China). This parameter presented a similar behaviour in many other countries, reaching a value at equilibrium of nearly a 5%. The high CFR values observed at the beginning of the epidemics in every country are probably due to several joint factors, including the weakness of the more vulnerable population (very old, already sick, people), or the lack of knowledge on which medical treatments were more effective. Those factors improved with time. In the cases of European countries a similar evolution was observed, although the decrease was slower than happened in China (at the beginning of April 2020 the CFR for USA was 0.3998, for Italy was 0.3557, and for Spain was 0.2052). The time delay, from the diagnose of an infected patient to the possible death, was adjusted at each country using real data. In the initial stages the observed delay was of 5 to 10 days for all the countries and reduced after some days.

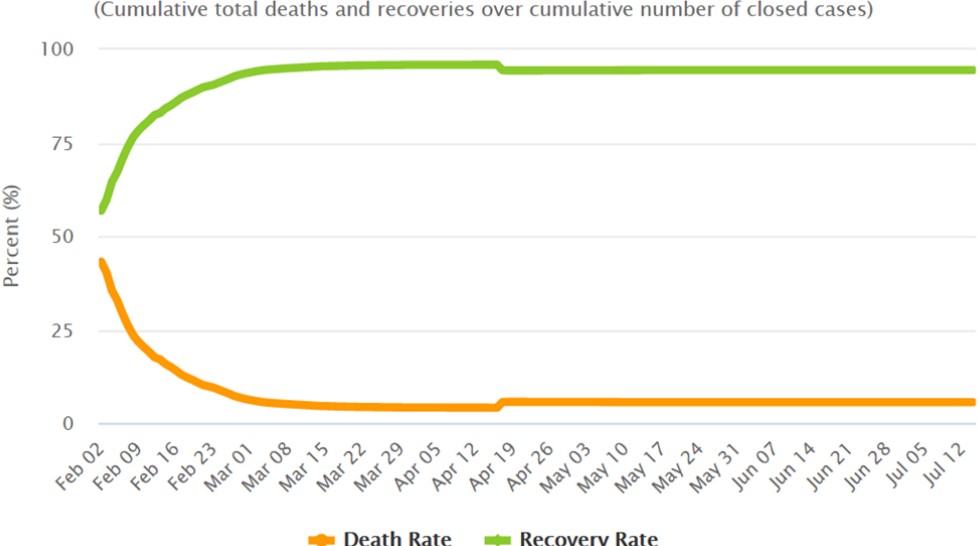

**Figure 4.** Case Fatality Rate as measured in China since the beginning of February 2020. The value measured the 4th of March of 5.78% was taken for the model (Source of information https://www.wo rldometers.info/coronavirus/ (consulted on 11 March)).

## 2.2. Parametrization during the Lock-Down

A non pharmaceutical intervention used in China and many other countries, included Spain, was the so called 'lock-down' in which the population is required to stay at home and only leave if essential. This NPI has been partially implemented in some regions and totally in others, including the region of Hubei in China (58.5 million inhabitants), Spain (46.6 million inhabitants) or Italy (60.4 million inhabitants).

In each region or country the initial value used for $N$ was its total population, but after the lock-down, the number of people already infected, or in contact with infectious people, is fixed and therefore $N$ would be smaller. This number cannot be determined before the lock-down but can be calculated the same day that the lock-down is implemented by using the number of infected measured at that exact time. A first estimation was made using the number of infected, estimated with the model, 14 days after beginning the lock-down (14 days was assumed to be the incubation period for the COVID-19) and multiplying that number by a factor of 10, which would provide the total infected. This method provides a rough estimation which needs further refinements when new data are obtained, however it provided valid estimations for forecasting the time when the maximum (peak) for daily new cases of diagnosed cases or deaths would be expected.

As expected, the daily infection rate $r$ was observed to decrease, from the rate the day before the NPI was applied (typically around 1.3) to a number slightly higher than 1.0, as observed at South Korea. The same behaviour was observed in every country and at every scale. After the lock-down is implemented, the $r$ parameter can be fitted by least squares to the curve given in Equation (2), for the given region or country.

$$r = 1 + A \cdot e^{-\alpha \cdot t}, \qquad (2)$$

where $t$ is the time in days since the lock-down and $A$ and $\alpha$ are constants empirically determined at the location. Table 3 shows some values for $A$ and $\alpha$ adjusted for different countries after a lock-down was implemented and examples of smaller scale regions within Spain (Andalusia and Catalonia were chosen for this example because they are the two more populated regions in Spain). Those values were obtained by fitting the Equation (2) to the experimental data in different regions or countries. (* Experimental data from worldometer (Source of information https://www.worldometers.info/coro

navirus/ (consulted on 17 April)). ** Experimental data from the Spanish official source of information (Source of information 'Instituto de Salud Carlos III' (ISCIII): https://covid19.isciii.es/)).

**Table 3.** Values obtained by fitting Equation (2) to the experimental data in different countries and two Spanish regions (Catalonia and Andalusia).

| Region | A | α |
|---|---|---|
| South Korea * | 0.226 | 0.235 |
| Italy * | 0.293 | 0.070 |
| UK * | 0.326 | 0.051 |
| Spain ** | 0.295 | 0.074 |
| Andalusia ** | 0.366 | 0.096 |
| Catalonia ** | 0.491 | 0.109 |

* (Source of information https://www.worldometers.info/coronavirus/ (consulted on 17 April)); ** ('Instituto de Salud Carlos III' (ISCIII): https://covid19.isciii.es/(consulted on 17 April)).

The number of individuals infected before the lock-down (*N*), and the constants *A* and *α* cannot be determined prior to the lock-down, as different groups of individuals or societies behave differently under the same exact government instructions, and also different governments provided slightly different instructions. So the only chance to obtain good predictions after the lock-down is to wait for several days to obtain experimental data to be used to fit the curve under the Equation (2). It should be also pointed out that, some sources of information provided data shifted in time or simply just different and consequently the fitting could provide different values for the parameters.

The parameters *H* and *C* were determined by averaging the empirical values from the studied region. In Spain the values obtained as an average up to the 6 April were $H = 0.467$ and $C = 0.1497$, which indicates that a high rate of diagnosed infected needed to be hospitalized, or more likely, that only severe cases were diagnosed at the hospitals, needing half of them to be admitted. Also a high percentage of the inpatients (almost a 15%) needed intensive care using ventilators, which was in agreement with the observed pattern in China and other countries. In this case $H \cdot C = 0.07$ (7%) which was in a good agreement with the initial range observed for the inpatients needing an ICU. As all parameters were dynamically calculated every day, the predictions were slightly calibrated daily.

Also for the CFR empirical values were used, as the equilibrium value taken from China was well surpassed in the initial stages in many European countries, although it tended to the same equilibrium value (nearly 5%). Although initially it reached values of even a 50%, the experimental CFR in Spain, as in UK, Belgium, or Italy, was 0.12 (12%) by April. The delay applied from reporting the positive diagnose of a patient to the death (when produced) was reduced to 3 days.

*2.3. Parametrization after Easing the Lock-Down*

When a region or a country decides to relax the confinement, the parameters need a new calibration to take into account the situation.

When the lock-down is completely abandoned *N* would return to be again the whole population of the region or country. However, this was not the situation in every country.

For example, in Spain the lock-down was established 15 March. Although the ideal situation would be to maintain the total lock-down until *r* reached a value close to 1, value expected to happen by the end of April according with the model, 13 April some relaxation was adopted, allowing most of the workers to return to their normal activities. A large part of the population remained confined, but a graded approach was established to remove it before the end of May. This being the situation, the parameters can be only inferred after some data are collected, following the same methodology established during the lock-down. Therefore *r* should be fitted to an exponential decrease, following the same Equation (2) after obtaining enough data. To perform initial conservative estimations a value of $r = 1.03$ can be used.

$$r = 1.01 + B \cdot e^{-\beta \cdot t}. \tag{3}$$

In the final stages of the easing of the lock-down, Equation (3) was used for *r*, considering impossible to achieve a value *r* < 1.01 (as the experience in other countries showed that reducing the level of daily infections below that value was, at least, very difficult). *B* and *β* are again constants empirically determined at the location

The rest of parameters: *H*, *C* or the CFR remain being averaged along the whole period with real data.

This parametrization can be used to assess the evolution of the situation after easing the lock-down or to design the strategy to optimize the number of infected, hospitalized or inpatients needing an ICU, to avoid the saturation of the health system of a country.

## 3. Results

As pointed out, 3 phases were considered:

1. An initial phase of the outbreak where no severe restrictions were applied.
2. A second phase where severe non pharmaceutical interventions (confinement) were applied.
3. A last phase where relaxation of the more severe NPI is assumed, although some keep being used.

As an example the application of the model with the appropriate parametrization in Spain is presented to show the performance of the model. The same methodology was also used for some of the regions in Spain, and can be used to any other region or country of any size.

### 3.1. Initial Phase

The initial phase is considered before any countermeasure is applied. Figure 5 shows the cumulated number of infected and the total number of deaths reported in Spain (in red) from 29 February/20 March. From 29 February/14 March reported data (in red) are shown against modelled values (in green). The schools closing was established from the 11th of March and the total lock-down the 15th of March.

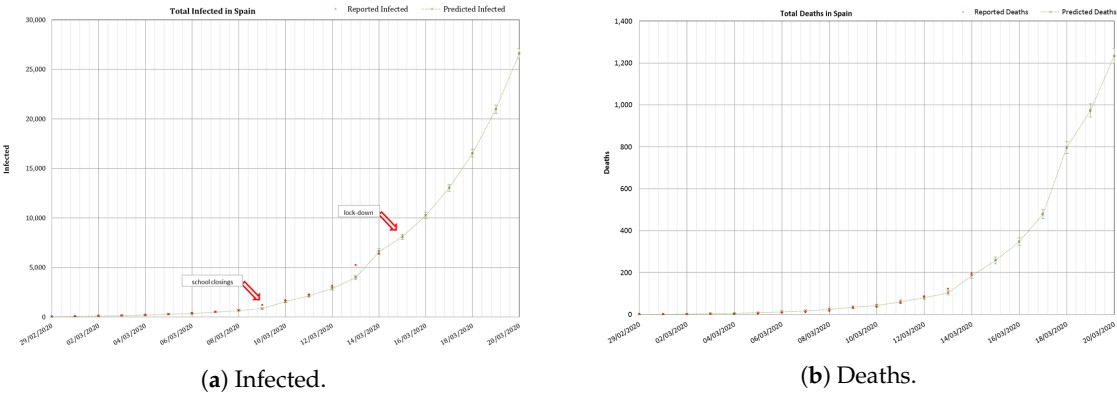

(**a**) Infected. (**b**) Deaths.

**Figure 5.** Predicted number of infected (**a**) and deaths (**b**) in Spain from 29 February/20 March. The data from the 14 March are based in the model assuming no interventions were implemented. Red dots—reported. Green bars—modelled with uncertainty.

As explained, during this initial stage, *r* = 1.05 was calculated as an average of the values measured during the initial days of spread of the epidemics; *N* was the total population in Spain ($46.6 \times 10^6$ inhabitants); and the $CFR = 0.0578$ was taken from the Chinese experience. In this phase the only parameter dynamically calibrated, to adjust the data reported daily, was the delay from the number of infected to the number of deaths, as was explained before, from an initial value of 7 days that was reduced up to a 2 days delay applied the 5th of March. In this specific case, the forecast indicated a number of infected cases of $26,600 \pm 500$ and a number of deaths of $1230 \pm 150$ to occur 6 days later (11th of March). The real number of reported infected was 21,571 (19% difference), and the number of reported deaths was of 1093 (11% difference). The number of inpatients needing an ICU

and a ventilator was calculated as $I(t) \cdot H(t) \cdot C(t)$, providing a range of [1330–3990]. The reported number of inpatients which needed an ICU the 20th of March was of 1630 (within the calculated range). Although in this initial stage many factors altered the real numbers the accuracy was reasonably good.

Predictions of the likely number of infected, hospitalized inpatients and total deaths were carried out using these conditions for the initial phase (uncontrolled spread of the disease). The model predicted that, if a severe NPI (total lock-down) was not adopted, but on the contrary the virus was left to spread without control, at the end of the epidemic in Spain 12 million people would have been infected, of which nearly 1 million people would need intensive care and about 700,000 infected would die directly because of the COVID-19 disease. However the number of deaths would likely be higher due to the saturation of the health system, as these numbers would occur in a very short period.

These results could provide an early idea of the urgent necessity of applying extreme NPIs like the total lock-down, they could be also used to predict the consequences of not applying the severe NPIs, and also to prepare for the capabilities of the ICUs, including the number of ventilators.

### 3.2. Lock-Down Phase

In Spain closing of schools began on 11 March and lock-down was established the 15 March. All factors were re-calibrated for this second phase as stated, including a fitting of the daily infection rate to the curve given in Equation (2). For the number of susceptible individuals $N$ an initial estimation ($N = 1.1 \times 10^6$) was carried out using the results of the model 14 days after the lock-down. An average value $r = 1.32$ was used. Results of these early predictions are presented in Figure 6. These values were later calibrated dynamically to $r = 1.20$ and $N \simeq 9 \times 10^6$ using data up to the 15th of March. Results of the predictions are presented in Figure 7.

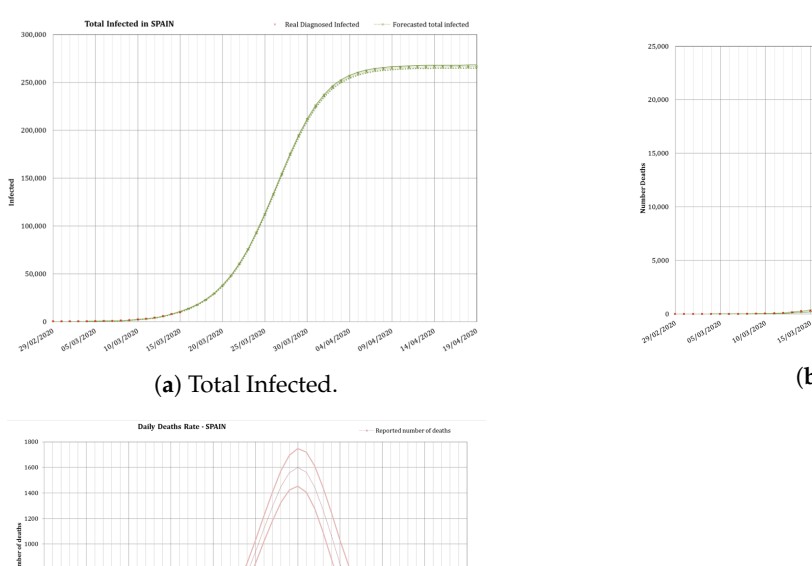

(a) Total Infected.

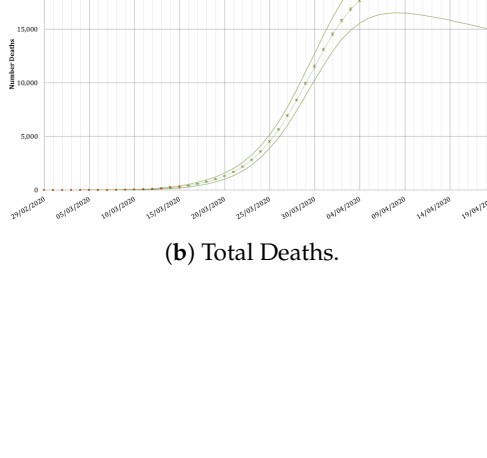

(b) Total Deaths.

(c) Daily Deaths.

**Figure 6.** Total number of infected (**a**), total deaths (**b**) and daily deaths (**c**) in Spain predicted from 29 February/19 April. The preliminary results assumed the total lock-down since the 15 March with data up to the 14 March. Red dots—reported data. Green bars—modelled values with uncertainty.

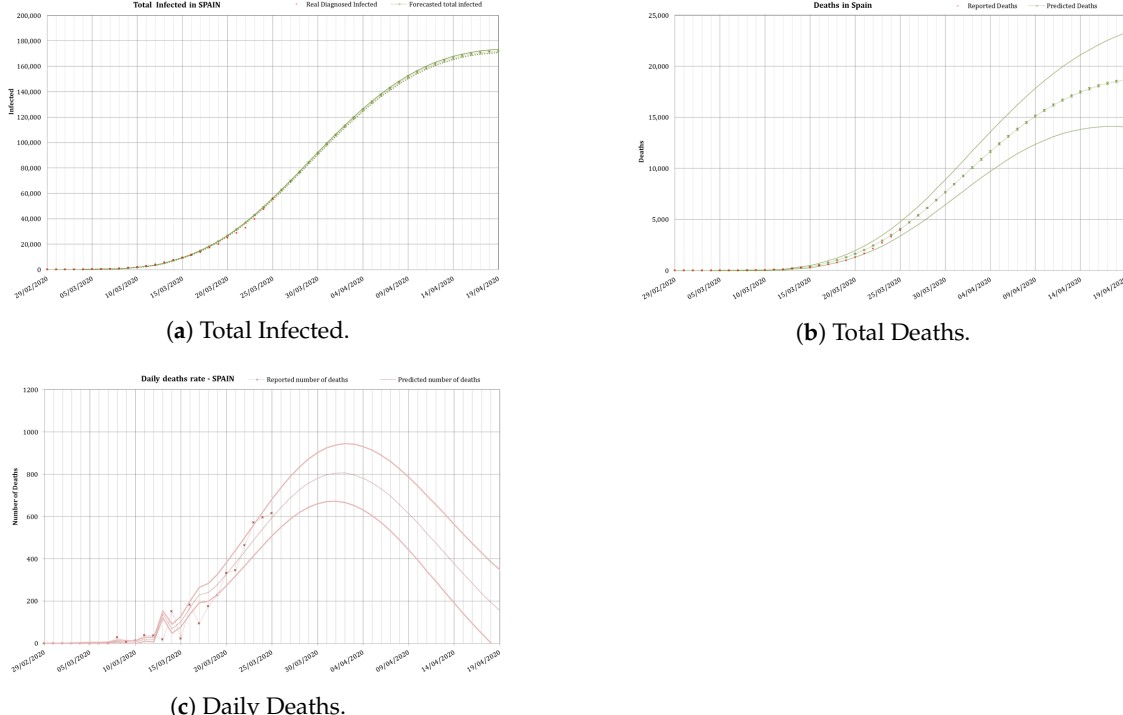

(**a**) Total Infected.

(**b**) Total Deaths.

(**c**) Daily Deaths.

**Figure 7.** Total number of infected (**a**), total deaths (**b**) and daily deaths (**c**) in Spain predicted from the 29 February/19 April. These second modelled results assumed the total lock-down since the 15 March and were calibrated with data up to the 25 March. Red dots—reported data. Green bars—modelled values with uncertainty.

The application of the total lock-down to the model reduced the predictions carried out the 26 March to a total of 194,000 infected, 85,700 hospitalized, nearly 8,600 with needs of an ICU and 19,500 ± 1400 deaths to occur by the 17 April. The real numbers reported at that date were 197,142 infected (1.6% difference), 7548 inpatients needing an ICU (12% difference) and 20,043 deaths (2.7% difference). The model predicted the peak for the rate of daily deaths to occur between the 29th of March and the 3rd of April. In reality the peak, after the administration revised the data (two months later), occurred the 31 March.

*3.3. Unlocking Phase*

The last phase is the ease of the lock-down. In this case, predictions based on some hypothesis, carried out during April, are presented in this paper and compared with the real evolution. Again the case of Spain is presented as example. Using the models presented in this paper recommendations were provided to ease the lock-down around 21 April [21]. In reality a partial unlock was decreed the 13th of April for non-essential workers, and a phased total unlocking from the 30 April, where some activities were allowed gradually each week until the 21 June, where normal activity was restored, although the population should follow NPI health countermeasures, as social distance, use of masks, washing of hands, and so forth.

After the unlocking many uncertainties appear, but the results of the model depend largely on the daily rate of infections $r$.

3.3.1. Partial Unlock

On the 13 April a partial ease of the total lock-down was applied in Spain.

In order to obtain conservative figures an initial forecast was carried out with the data available the 16 April [22]. At that point only some reasonable hypotheses could be applied to calibrate $N$ and $r$.

For $N$ it was assumed that about a 20% of the total workforce (in Spain 20 million workers on 2020) went back to work, as only some industries were allowed to begin again their activities, after that day. As those people could also infect their families, 2 further members on average, an initial $N = 1.2 \times 10^7$ was taken. In order to carry out conservative predictions $r = 1.03$ (a daily increase of the infected of a 3%) was taken. Results in Figure 8 were obtained for those conservative assumptions.

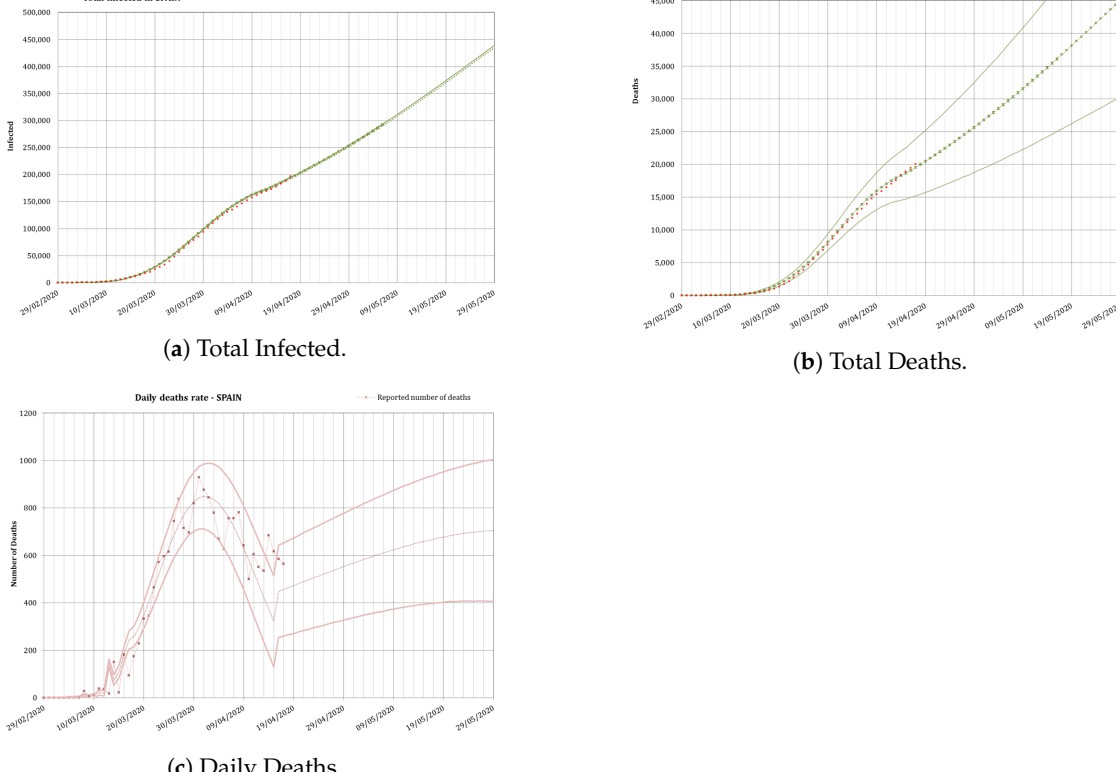

(**a**) Total Infected.

(**b**) Total Deaths.

(**c**) Daily Deaths.

**Figure 8.** Total number of infected (**a**), total deaths (**b**) and daily deaths (**c**) in Spain predicted from 29 February/29 May. These modelled results assumed the ease of the lock-down since the 13 April with conservative assumptions for $N$ and $r$. Red dots—reported data. Green bars—modelled values with uncertainty.

Using these conservative values of $N$ and $r$, some consequences in the partial ease of the lock-down could be extracted. First of all, the number of diagnosed infected people would increase continuously beyond May. In fact there would been a peak in the daily rate of infected by the 28th of May and a peak in the daily rate of deaths by the 1st of June. The total number of deaths in Spain by the end of April would reach the $46,000 \pm 15,000$ under this scenario. That was the conservative value we published in April [22]. If this would have been the case ($r > 1.03$) a saturation of the health system would have occurred again in Spain. So that value of $r$ could be regarded as an upper bound which should be avoided.

In reality, a very good behaviour of the Spanish population made $r$ to continuously reduce even after a total easing of the lock-down.

### 3.3.2. Total Unlocking

From the 4th of May a total unlock was applied in Spain, with a progressive increase in the mobility of the people since then, and therefore a recalibration was needed. For this phase $N$ was again considered the total population (46.6 millions inhabitants).

Obtaining some more data a least squares fit was performed using an initial daily infection rate $r = 1.02$ (2% daily increase of infected) the day before the unlocking, and the Equation (3) to fit $r$.

The calibration resulted in $B = 0.03$ and $\beta = 0.153$, what gives the equation $r = 1.01 + 0.03 \cdot e^{(-0.153 \cdot t)}$. The results obtained with this fit is shown in Figure 9.

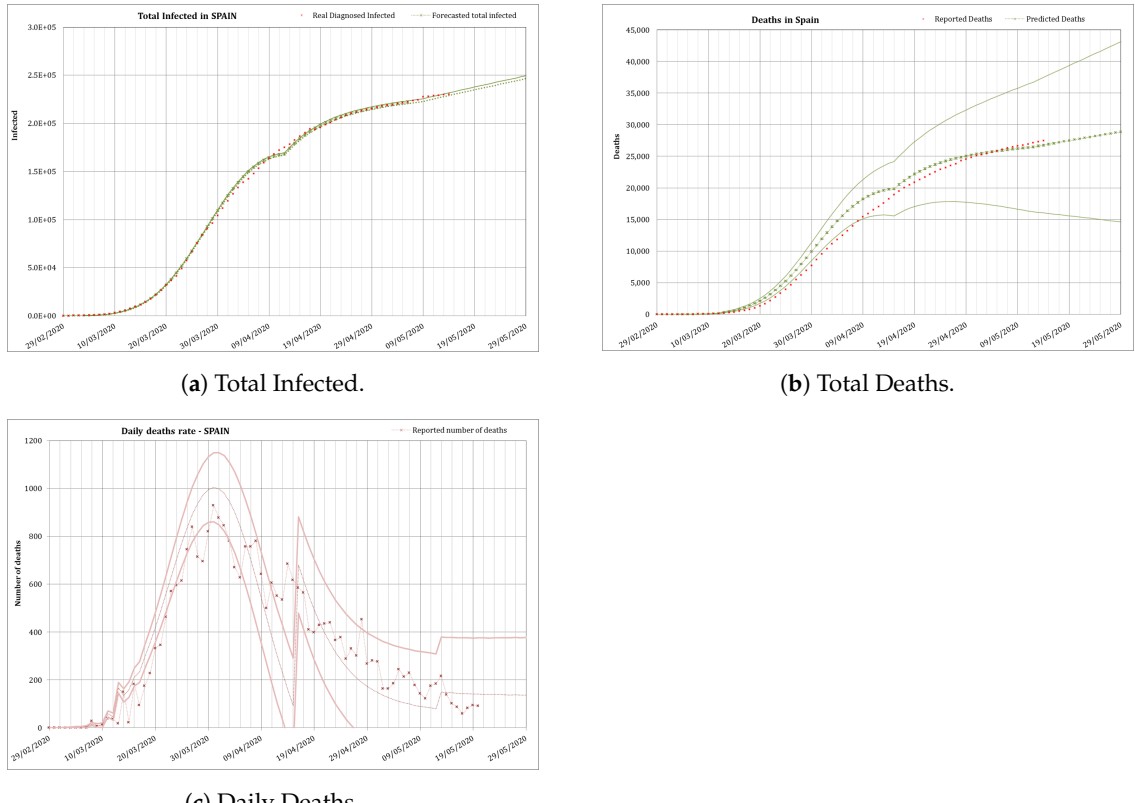

(**a**) Total Infected.

(**b**) Total Deaths.

(**c**) Daily Deaths.

**Figure 9.** Total number of infected (**a**), total deaths (**b**) and daily deaths (**c**) in Spain predicted from 29 February/28 June. These modelled results assumed the ease of the lock-down since the 13 April. Red dots—reported data. Green bars—modelled values with uncertainties.

The series of data finishes at the end of May, as official aggregated data for Spain were no longer provided. In fact least squares fit was extremely difficult as there was an attempt to homogenize the data between the different regions in the country which made the whole series of data to be revised almost every day. That is one of the reasons why in this occasion the fitting was not as good as in previous phases for the total number of infected and the total number of deaths, as the focus was put in the daily number of deaths to obtain a good fitting. Daily number of deaths was the main endpoint in this phase because this indicator is the best one to perform future surveys of the situation of the epidemic.

The results obtained using this calibration for this final stage was that the number of diagnosed infected people would slowly increase continuously beyond May. This is a logical result, as the infection would be always present in a slow rate until the virus is eradicated or there is a vaccine to control the spread of the disease. The predicted number of total infected is of $317,500 \pm 1700$ and the total number of deaths would be 37,100, with a huge uncertainty, by the 1 August. The real numbers that day were 335,602 reported infected (5.7% difference) and 28,445 deaths (23% difference).

During the summer the situation was controlled, however if the good practices in the application of NPIs are abandoned: hands washing, social distance, use of masks, and so forth, $r$ could easily reach values above 1.03, surely another increase in the number of infected will occur. This will be the case also when borders are reopened and new infectious people (even asymptomatic) enter inadvertently from countries in the initial phase of the epidemics.

Of course, these predictions did not consider important changes, as the discovery of a vaccine—which seems extremely difficult in a short period of time—or the increase in the

temperatures in the summer which could reduce the infectivity of the SARS-CoV-2, or a higher isolation which could reduce the severity of the COVID-19 due to a higher production of vitamin D [29,30], or any other unforeseen circumstance. During that summer time, of course, the treatment of hospitalized patients has improved and therefore a smaller fraction of inpatients need an ICU or even die.

### 3.4. Percentage of Infected Population in the Regions

There was an additional result extracted from this model. The need to recalibrate it during the locking phase by fitting the parameter $N$, the number of people susceptible to be infected in that phase, offers the possibility of using that number to infer the percentage of the population in a country or region which could have been infected, for this particular virus, most of them showing no symptoms.

As an example these numbers were extracted for the autonomous communities (administrative regions) in Spain and transformed to three levels of infection (low—below 5% —, medium—from 5% to 10%— and high—above 10%), giving rise to the result shown in Figure 10a.

This qualitative result has an important use for the authorities, as the population which have a high level of infection by the SARS-CoV-2 did probably developed immunity against the virus (at least temporarily as discussed before), and therefore there is no chance for them to be infected again in the close future. And on the contrary, there is a bigger chance of developing further outbreaks of the disease in those regions where the percentage of the infected population was smaller.

The model provided some counter-intuitive results. For example, the capital of Spain, Madrid, presented "medium level" of infection of the population, whereas Catalonia showed "high level", while both, the number of diagnosed cases and the number of deaths in Madrid was higher, and this would imply a higher level of infection in Madrid. However this could be explained in the different criteria followed by the different regions for reporting the numbers. For example Catalonia decided by mid of May to include in the statistics the deaths of people occurred out of the hospitals, while it was unknown if Madrid was including those deaths already in the statistics. The same occurs with the number of diagnosed infected cases, as there is observed an increase in the number of tests performed on people who finally did not need a hospital in Madrid, the percentage in Catalonia of diagnosed population finally needing a hospital was still around 65% by May.

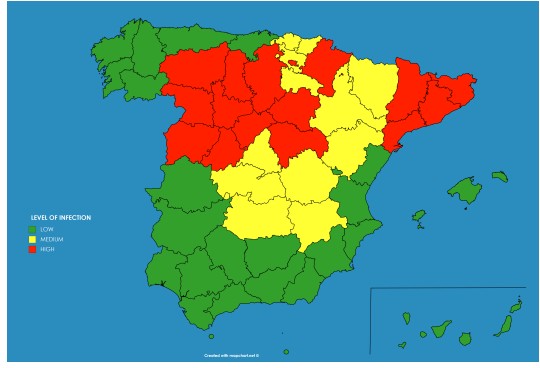
(**a**) Forecasted levels of infection [22].

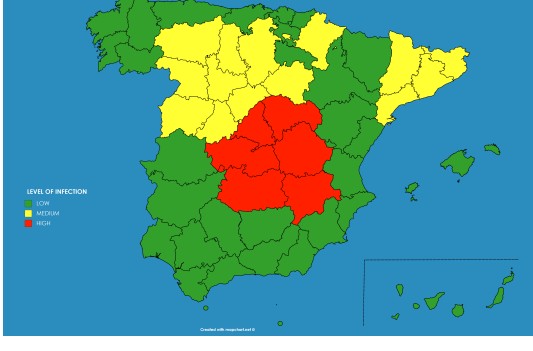
(**b**) Measured levels of infection [31].

**Figure 10.** Levels of infection in the population at each region of Spain: calculated in April 2020 by using the model presented in this paper (**a**) and measured at the seroprevalence study ENE-COVID finished in July 2020 (**b**).

These predictions were compared against the extensive statistical program of immunity prevalence carried out on the Spanish population from April to June 2020, published in early July [31] (see Figure 10b). This study measured the percentage of people infected during the epidemic, taking account of asymptomatic infected persons, while the previous reported numbers included just the

hospitalized inpatients showing severe symptoms on which a polymerase chain reaction (PCR) test returned a positive result. The results of both studies can be compared in Figure 10.

As can be seen the results provided by the model in April were accurate in most of the regions were low infection occurred, as in the south (Andalucia, Extremadura, Ceuta, Melilla, Canary Islands, Balears Islands, Valencia and Murcia) or north-west (Galicia, Asturias and Cantabria) of Spain. However the model predicted a medium to high infection level at the whole north and north-east, whereas the measured levels of seroprevalence obtained low levels at some regions (La Rioja, Navarra and Basque Country). In general the forecast provided a good general view of the infection level.

## 4. Discussion and Conclusions

As it is unlikely that a vaccine to the SARS-CoV-2 or a cure of the associated disease COVID-19 is developed in the next months, the only way of reducing the consequences of the epidemic at this moment is an optimum application of the available NPIs.

Traditional epidemiological models like compartment models need good quality data to offer proper predictions. However, during the outbreak of an epidemic often the quality of the data is not the primary concern of the health systems and therefore many epidemiologists were discouraged and abandoned their efforts of offering predictions. This fact, together with the need of fewer data, made it interesting using a semiempirical model based in the logistic map. The application of such methodology to the different phases of the COVID-19 epidemics in Spain was shown. This methodology provided good results in the forecast of the evolution of the disease in every situation.

The use of extreme non-pharmaceutical interventions, such as the total lock-down, showed their effectiveness during the period they were applied. However, easing the countermeasures allow new outbreaks of the infection to appear. This situation forces the need of applying many simultaneous techniques to reduce the effect of the disease if that is the case. One of those techniques could be the application of the methodology described in this paper to provide early alerts of the outbreaks in countries or smaller units of population, allowing an optimization of sanitary resources and reducing the economic and social impact of future NPIs applied locally.

All the data used in the paper were data officially published in real time by the official sources from the Spanish Government [32,33]. As was shown, reasonably accurate results can be produced by using the model presented in this paper to the different phases of an epidemic. In a previous preprint, assuming an infection daily rate $r$ of 3%, a total number of 400 000 diagnosed infected and a total number of 46,000 ± 15,000 deaths were forecasted in Spain by the end of May [22]. Those predictions overestimated the real values due to a more strict reduction of the infection daily rate in the country, reaching values below 1%.

The forecasts covered from the number of infected, hospitalized, inpatients needing an ICU or deaths, to the time where the peak of daily deaths would be produced or the level of infection in a given region. In the last prediction, carried out for the beginning of August, 317,500 ± 1700 infected and a total number of deaths of 37,100 were predicted, with a huge uncertainty, to be compared with the real numbers of 335,602 reported infected (5.7% difference) and 28,445 deaths (23% difference).

The aim of any policy dealing with the application and withdrawal of NPI should carefully consider daily infection rates. In the case of the COVID-19 a daily infection rate $r$ lying within the range of 1.01 to 1.02 (1% to 2% daily increase), as was shown in countries like South Korea, would produce a manageable level of people needing an ICU in hospitals, avoiding the saturation of national healthcare systems and therefore unnecessary deaths.

Also a qualitative prediction of the percentage of the population infected in the different regions of Spain was performed by using the suggested semi-empirical model. These predictions were compared against the extensive statistical program of immunity prevalence carried out on the Spanish population from April to June 2020, published at early July, showing that the model provided in April reasonable results in most of the regions, although the model predicted a medium to high infection level at the whole north and north-east, while the measured levels of seroprevalence obtained low levels.

Some results obtained with this methodology were not intuitive according to the official information. The more counter-intuitive result probably being the higher level of infection of Catalonia compared with Madrid region. As said, in general the forecast provided a good forecast of the infection level.

The COVID-19 epidemic is still ongoing and the knowledge will increase with time. In the next future new outbreaks are foreseen in the countries where the first one was controlled, unless a vaccine or a cure are developed in the next future. Therefore models will be needed to forecast again the evolution and to advice the authorities in the needs of the country's health system. Some characteristics of the virus, needed to perform better predictions, are still unknown, as the lost of immunity of cured individuals or the influence of vitamin D in the severity of the disease.

A continuous watch of the disease is still needed to provide proper advice which can be used by policy makers.

**Author Contributions:** Conceptualization, Data Curation, Methodology, Software and Writing by J.C.M.; Validation, Writing—review and editing and Formal analysis by S.P. and A.D. All authors have read and agreed to the published version of the manuscript.

**Funding:** This research received no external funding.

**Acknowledgments:** The authors would like to acknowledge to all the colleagues who have constructively read and discussed the paper to improve its content. Specially to Ignacio Rodríguez and Asunción Núñez from Dragonfly Innovation Technologies, UK; Justin Smith, from the Public Health England, UK; and Alejandro Ubeda from the Ramon y Cajal Hospital, Spain.

**Conflicts of Interest:** The authors declare no conflict of interest.

## Abbreviations

The following abbreviations are used in this manuscript:

| | |
|---|---|
| CoV | Coronavirus |
| COVID19 | Coronavirus Disease 2019 |
| CFR | Case Fatality Rate |
| DOAJ | Directory of Open Access Journals |
| ICU | Intensive Care Units |
| MDPI | Multidisciplinary Digital Publishing Institute |
| NPI | Non Pharmaceutical Interventions |
| SARS | Severe Acute Respiratory Syndrome |
| SEIR | Susceptible Exposed Infectious Removed |
| SEIRS | Susceptible, Exposed, Infectious, Removed and Susceptible |
| SIR | Susceptible Infectious Removed |
| WHO | World Health Organization |

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
