# Peer review of "Application of a Semi-Empirical Dynamic Model to Forecast the Propagation of the COVID-19 Epidemics in Spain"

_forecasting, doi:10.3390/forecast2040024_

Round 1
Reviewer 1 Report
The paper proposes a parametric approach to forecast the COVID-19, using the Spain case as an example. The paper seems good but lacks on a literature review.
My suggestions are:
1) Please provide a full literature review about epidemiological, statistical, and machine learning approaches used to forecasting COVID-19 worldwide. Also, provide a critical analysis of these papers.
2) Clarify to the reader why the authors choose a semi-empirical model and their differences regarding some classical approaches such as the SIR or SEIR model.
3) Please, add in the introduction the contributions and objectives of the paper.
4) Improves the quality of Figure 3. Moreover, it seems that the figures were obtained in different software. Please, use the same software for all figures.
5) Finally, the discussion lacks literature results that support the paper results. It must be improved.
Reviewer 2 Report
Authors presenting semi-empirical model approaches in prediction of COVID-19 epidemic size in Spain. The description and significance of the study are well promised to achieve remarkable outcomes. However, I would like to suggest some necessary changes that can explore the article's importance.
The current trend, and spread of COVID-19 underpins the complexity of the global pandemic. Hence, a current study has showed how confirmed cases, even across cities may be affected by common shocks.
the authors could justify with literature the measures they propose to stop the advance of the pandemic and recent works on mathematical models for COVID-19 has to be cited
In materials and methods, Using generalize statement in R0 assumptions cannot establish causality. I suggest taking a cue in line with the finding and empirical interpretations presented in "COVID-19 outbreak reproduction number estimations and forecasting in Marche, Italy"
Figure 4 resource should mention (I am wondering authors include this graph by adoption from world meter site).
How was quality control assured in the modelling? How was it validated?
Round 2
Reviewer 1 Report
Dear Editor, after this new version of the manuscript, I recommend it for publication.
Reviewer 2 Report
The authors successfully presented the response for given comments.